# High-Fat Diet-Induced Dysregulation of Immune Cells Correlates with Macrophage Phenotypes and Chronic Inflammation in Adipose Tissue

**DOI:** 10.3390/cells11081327

**Published:** 2022-04-13

**Authors:** Sonia Kiran, Ahmed Rakib, Sunitha Kodidela, Santosh Kumar, Udai P. Singh

**Affiliations:** Department of Pharmaceutical Sciences, College of Pharmacy, The University of Tennessee Health Science Center (UTHSC), 881 Madison Avenue, Memphis, TN 38103, USA; skiran@uthsc.edu (S.K.); arakib@uthsc.edu (A.R.); skodidel@uthsc.edu (S.K.); ksantosh@uthsc.edu (S.K.)

**Keywords:** obesity, inflammation, Th17, Tregs, macrophages

## Abstract

Obesity is a complex disease associated with various metabolic abnormalities, cardiovascular diseases, and low-grade chronic inflammation. Inflammation associated with T helper 1 (Th1) immune cells is dominant in adipose tissue (AT) and exerts metabolically deleterious impacts. The precise mechanism of alteration in AT immune system and its effect on metabolic homeostasis remains unclear. In this study, we investigated how a high-fat diet (HFD) alters the AT immune response and influences inflammation during obesity. HFD consumption amends the metabolic parameters, including body weight, glucose, and insulin levels. We observed increased infiltration of Th17 cells, a subset of dendritic cells (CD103^+^), and M1 macrophages in AT of mice fed HFD compared to those fed a normal diet (ND). In mice that were fed HFD, we also observed a reduction in regulatory T cells (Tregs) relative to the numbers of these cells in mice fed ND. Corresponding with this, mice in the HFD group exhibited higher levels of proinflammatory cytokines and chemokines than those in the ND group. We also observed alterations in signaling pathways, including increased protein expression of IRF3, TGFβ1, and mRNA expression of IL-6, KLF4, and STAT3 in the AT of the mice fed HFD as compared to those fed ND. Further, HFD-fed mice exhibited decreased protein expression of peroxisome proliferator-activated receptor-gamma (PPAR-γ) compared to mice fed ND, suggesting that PPAR-γ functions as a negative regulator of Th17 cell differentiation. These results suggest that HFD induces increased levels of inflammatory cytokines and key immune cells, including Th17, M1 macrophages, and CD103^+^ dendritic cells, and reduces levels of PPAR-γ and Tregs to sustain AT inflammation. This study supports the notion that dysregulation of Th17/Tregs, which polarizes macrophages towards M1 phenotypes in part through TGFβ1-IRF3-STAT3 and negatively regulates PPAR-γ mediated pathways, results in AT inflammation during obesity.

## 1. Introduction

Obesity is a major public health concern that has reached pandemic proportions. Currently, 42% of the American adult population is obese or overweight [1,2]. Obesity is also high in young children and adolescents [3]. Obesity in both mice and humans is associated with chronic low-grade inflammation within adipose tissue (AT), which plays a key role in the development of pathological processes associated with obesity, including type 2 diabetes (T2D) [4,5]. Thus, AT inflammation has emerged as a crucial process that links obesity with its associated pathology. AT inflammation in obese patients is characterized by a significant accumulation of immune cells that express proinflammatory cytokines and chemokines [6,7]. CD4^+^ and CD8^+^ T cells serve as critical regulators of chronic inflammation and participate in abnormal energy metabolism [8]. The crosstalk between immune cells and adipocytes that encompasses the early triggers and signals for maintenance of AT inflammation in obesity remains elusive, limiting our ability to develop novel therapeutics to treat it. Thus, a better understanding of the intricate balance between T helper 17 (Th17) cells and regulatory T cells (Tregs) in AT that leads to switching of macrophages to predominantly the M1 phenotype, will help to identify the signaling pathways that establish a norm to maintain obesity and metabolic syndrome. These pathways will provide targets for the development of drugs that effectively prevent or suppress obesity and metabolic syndrome. The relationship between obesity, chronic inflammation, and metabolic syndrome remains unclear to date. Thus, the results from this study could represent a unique approach to managing both obesity and AT inflammation.

During obesity, the presence of excess lipids and glucose can directly impact the differentiation and function of Th17 cells and Tregs via nutrient sensors. Th17 cells are associated with obesity-induced inflammation [9]. The proportion of circulating Th17 cells and IL-17 levels were significantly higher in obese children than in a non-obese control group and were positively correlated with body mass index (BMI) [10,11]. Obese individuals and mice exhibit increased infiltration into AT Th17 cells [12], the phenotype and function of which promote AT inflammation and many other autoimmune diseases [13]. Alterations in Th17/Treg function are observed in obese visceral adipose tissue (VAT) and insulin-resistant conditions, compared to lean controls [14,15]. Similarly, a Th17/Tregs imbalance is observed in the VAT of obese and T2D patients; however, a decreased number of Tregs defend against obesity-associated metabolic syndrome and inversely correlate with BMI [16,17,18]. Since the imbalance of Th17/Tregs plays such a fundamental role in obesity-dependent inflammation by mechanisms that remain poorly defined, a better understanding of how the ratio of Th17 cells and Tregs mediate obesity and metabolic disorders will be needed to effectively prevent or suppress these conditions.

Macrophages are the main culprit in most cases of AT inflammation, and alteration of their phenotypes can support the development of inflammation, obesity, and related metabolic disorders [19]. In both humans experiencing obesity and mice with HFD-induced obesity, the massive accumulation of macrophages in fat depots is accompanied by a shift in the ratio of macrophages of the M2 (anti-inflammatory) to M1 (pro-inflammatory) phenotype [20,21,22]. During obesity, the numbers of M1 macrophages in AT increase and correlate with AT inflammation and insulin resistance (IR) [23]. In lean humans and mice, M2 macrophages predominate, secrete anti-inflammatory cytokines, and utilize oxidative metabolism to maintain AT homeostasis [24]. Interestingly, effector T cells (Teff) recruit macrophages during AT inflammation [25]. Dendritic cells (DCs) and a specific subset of macrophages that express a low level of CD11c and are recruited to AT during diet-induced obesity are key participants in the innate and adaptive immune response [26]. DCs also accumulate in the AT of both mice and humans in proportion to the BMI. Specific subsets of DCs (inflammatory DCs) that express macrophage markers CD11b/CD11c in humans or CD11c^high^F4/80^low^ in obese mice induce Th17 differentiation and accumulation [27]. Therefore, a better understanding of this process is required to evaluate whether the expression of Th17 by DCs is key to macrophage function in AT and whether Th17 can act as a therapeutic marker to suppress or completely prevent obesity and its related inflammation.

The polarization of macrophages towards the M1 phenotype, activation of Th17, the existence of an increased subset of DCs in AT that express macrophage markers, and enhanced levels of proinflammatory cytokines all are linked with AT inflammation. How HFD induction alters the Th17/Tregs ratio in AT and mediates the recruitment, polarization, and function of macrophages during obesity remains unclear. Thus, to understand this, we hypothesize that HFD induces inflammations in AT by dysregulating Th17 and Treg subsets to polarize macrophages towards M1 phenotypes to sustain chronic inflammation. In this study, we used a murine model of HFD-induced obesity to ascertain the link between Th17/Tregs dysregulation and DCs recruitment, which might lead to alteration of the macrophage phenotypes and functions in the AT microenvironment. Taken together, our results suggest that HFD induces Th17 cells, CD103^+^DCs, and inflammatory cytokines and reduces Tregs, which correlates well with TGFβ1, IRF3, STAT3, KLF4, and PPAR-γ mediated signaling pathways that might polarize macrophages toward M1 phenotypes to maintain low-grade AT inflammation. This study supports the notion that Th17/Tregs dysregulation plays a key role in altering M1 macrophages and inducing AT inflammation during obesity.

## 2. Materials and Methods

### 2.1. Animals

Wild-type (WT) male C57BL/6 mice (aged 7 weeks) were purchased from Jackson Laboratories (Bar Harbor, ME) and housed in an animal facility at the University of Tennessee Health Science Center Memphis (UTHSC). All experiments were conducted according to age and gender-matched mice under an approved protocol. It has been shown that male mice gain more weight and produce more disease-promoting immune cells as compared to females; this was the reason to consider male mice in this study [28]. However, in other ongoing functional studies, we have used both genders of mice. The mice were housed in isolator cages under conventional conditions for an initial one week of acclimatization to the animal facility using a normal 12:12 h light and dark cycle. All animal experimentation was performed under a protocol number (US-20-0162) approved by the UTHSC Institutional Animal Care and Use Committee (IACUC). All necessary precautions were taken during usage for various experimental procedures, especially to minimize undue discomfort and pain of the mice. Our power analysis suggested that six mice per group were appropriate to allow uncovering the effect sizes and standard deviations between experimental groups at a significance level of *p* < 0.05; 5% using two-sided *t*-tests, conducted separately or within the context of ANOVA. The experiments in this study were repeated three times to achieve statistical relevance and for a prudent conclusion.

### 2.2. Induction of Obesity in Mice by HFD

After one week of acclimatization in the animal facility, mice were randomly split into two experimental groups consisting of six mice each. At 8 weeks of age, one group of animals was switched to a normal diet (ND) containing 10% fat (*n* = 6; D12450J, Research Diets, New Brunswick, NJ, USA) to serve as the control group, while the other half were switched to a high-fat diet containing 60% fat (HFD; *n* = 6; D12492, Research Diets). Both HFD and ND were calorie-matched diets. In brief, the fat content in HFD (D12492) used ~54% lard and 68.8 gm % sucrose and matched the ND (D12450J) formulated with ~54% corn starch and 68.8 gm % sucrose in such a way that animals in ND also consumed the same amounts of calories as well as the same amounts of sucrose. Each diet was continued until the experimental endpoint for 12 weeks. We observed the mice every day for their general health and consistency in stool to assess the onset of diarrhea or other behavioral symptoms. The body weight of each mouse was measured every week while normal blood glucose levels without fasting were examined at the start and experimental endpoint of the studies. Body composition was also assessed using Echo MRI at the UTHSC animal core facility. After 12 weeks on the respective diets, mice were euthanized at the experimental endpoint by an overdose of isoflurane vapor. The spleen, mesenteric lymph nodes (MLNs), and epididymal adipose tissues (AT) were harvested from each mouse to prepare single-cell suspensions for later studies; blood was collected from the venous sinus of each mouse with the help of a capillary tube purchased from Universal Medical (Oldsmar, FL, USA).

### 2.3. Preparation of Single-Cell Suspensions

The spleen and MLNs were dissociated using a stomacher (Seward Stomacher^®^ 80) to make single-cell suspensions. The red blood cells were removed from whole blood with the help of a lysis buffer (Thermo fisher scientific Carlsbad, CA, USA), and cell debris was removed by using a cell strainer with a 70-micron (μ) filter. Cells were washed with RPMI 1640 media containing 10% fetal bovine serum (FBS) and stored at 4 °C for flow cytometry staining within 4 h of cell isolation, as described below. Epididymal AT was collected from peripheral fat pads and gently placed in MACS C tubes (MACS Miltenyi Biotec, 130-096-334 Auburn, CA, USA), processed using an adipose tissue dissociation kit (MACS Miltenyi Biotec, 130-105-808) with the help of a gentle MACS™ Dissociator (MACS Miltenyi Biotec, 130-093-235), and analyzed by RT-qPCR as described below. All isolated cells from the spleen, MLNs, and epididymal AT were incubated with 1 μg/mL ionomycin and 100 ng/Phorbol 12-myristate 13-acetate (PMA) (Sigma-Aldrich St. Louis, MO, USA) for 6 h followed by flow cytometry staining and analysis of Th17 cells.

### 2.4. Measurement of Cytokines and Chemokines

The serum concentrations of cytokines and chemokines including interleukins IL-5, IL-6, IL-10, IL-13, and IL-17, tumor necrosis factor-alpha (TNF-α), regulated on activation, normal T cell expressed and presumably secreted (RANTES), granulocyte colony-stimulating factor (G-CSF), macrophage inflammatory protein 1 beta (MIP1β), and interferon-gamma (γ)-induced protein 10 kDa (IP-10), also known as C-X-C motif chemokine 10 (CXCL10), were examined by using a Luminex-based multiplex ELISA assay kit (Bio-Rad, Hercules, CA, USA). In brief, the assay buffer of this kit contains a multiplex mixture of fluorophores that are conjugated to antibodies specific for mouse cytokines and chemokines IL-5, IL-6, IL-10, IL-17, TNF-α, G-CSF, RANTES, MIP1β, and CXCL10 analyte beads. First, these beads were added to pre-wet filter-bottom wells containing 50 µL of assay buffer. After the washing step and removal of the buffer, we added 50 µL of standard or mouse serum samples acquired from mice fed ND or HFD to each well, and the plate was incubated at 4 °C overnight with continuous gentle shaking. The next day, plates were washed by washing buffer three times. Next, we added 25 µL of biotinylated anti-mouse detection antibodies (Abs) specific for separate epitopes on the aforementioned proteins after 30 s vortexed at 300× *g* to each well, and the plates were incubated at room temperature (RT) for 30 min. Next, 50 µL of the streptavidin-phycoerythrin (streptavidin-PE) detection solution was added to each well, and the plate was incubated with constant shaking for 10 min at RT. After washing three times, we added 125 µL of assay buffer to each well, and signals were detected with a Bio-Plex instrument using Luminex™ xMAP technology (Bio-Rad Laboratories, Austin, TX, USA). Data analysis was performed using Bio-Plex Manager software (BioRad). This BioRad™ xMAP assay can detect >10 pg/mL for each analyte as suggested by the kit manufacturers.

### 2.5. Flow Cytometry Analysis

Compensation beads, isotype controls, and fluorescent-conjugated antibodies were purchased from BD Bioscience (San Diego, CA, USA) and Biolegend (San Diego, CA, USA). For each experimental group, freshly isolated cells (1 × 10^6^/mL) were pooled, pelleted, and resuspended in 80 µL of ice-cold flow cytometry staining buffer (PBS containing 1% FBS). The cells were stained with 5 µL of the manufacturers’ recommended dilutions of fluorescent-conjugated antibodies or their respective controls at 4 °C for 40 min. We used the following fluorescently labeled mouse monoclonal antibodies for flow cytometry: anti-CD11b (clone M1/70), anti-CD 11c (clone N418), anti-GR-1 (clone RB6-8C5), anti-CD103 (clone 2E7), anti-CD4 (clone GK 1.5), anti-FoxP3 (clone MF-14), anti-IL-17 (clone TC11-18H10-1), and anti-IFN-γ (clone XMG1.2). For intracellular staining of IL-17 and Tregs (Foxp3), the cells were washed twice with flow staining buffer and thoroughly re-suspended in BD Cytofix/Cytoperm solution (BD-PharMingen, San Diego, CA, USA) for 20 min. The cells were again washed twice with BD perm/wash solution after storage at 4 °C for 10 min. Intracellular staining and analysis of IL-17 and Foxp3 (Tregs) expression were performed according to Biolegend’s recommended protocol at RT for 30 min. Afterward, the cells were washed twice with flow cytometry staining buffer and resuspended in 300 µL of staining buffer. An Agilent flow cytometer (Novocyte) was used to quantify the fluorescent signals, which were expressed relative to the isotype control.

### 2.6. RT-qPCR Analysis

Total RNA was extracted from AT using a miRNeasy Mini Kit (QIAGEN, Hilden, Germany). Briefly, 100 mg of AT was homogenized in 700 µL of TRIzol lysis reagent and 140 µL of chloroform. After centrifugation, RNA was extracted with chloroform, precipitated with isopropyl alcohol, resuspended in 30 μL of RNA-free water, and the concentration and purity of each RNA sample were quantified using UV absorbance at 260 nm on a Nanodrop 2000c Spectrophotometer (Thermo Fisher Scientific, Wilmington, DE, USA). One microgram of extracted RNA in each sample was reverse-transcribed to cDNA using an iScript cDNA synthesis kit (Bio-Rad) according to the manufacturer’s procedure. The mRNA expression of targeted genes including GAPDH, KLF4, IL-6, and Stat 3 was measured by quantitative PCR (qPCR) using the appropriate primers and iTaq Universal SYBR Green Supermix (Bio-Rad) with a CFX96 Touch™ Real-Time PCR Detection System (Bio-Rad). Each reaction contained 1 μL of cDNA and 200 nM of each primer in 20 μL total volume. Thermocycling consisted of an initial hot start cycle (95 °C for 30 s), followed by 40 cycles at 95 °C for 5 s, 60 °C for 15 s, and 72 °C for 10 s. Primers were obtained from Qiagen Inc. Germantown MD (mouse Stat3: LPM17900A-200; mouse IL6: PPM03015A-200 and mouse KLF4: PPM25088B-200).

### 2.7. Immunoblot Analysis

AT-derived immune cells from mice that were fed either ND or HFD were pooled, lysed, and the total protein concentration of each pool was estimated using bicinchoninic acid (BCA) assay. A total of 30 µg of protein from each pool was separated on a 10% sodium dodecyl sulfate (SDS) gel and transferred to polyvinylidene difluoride (PVDF) membranes using a Transblot turbo instrument (Bio-Rad). The membrane was blocked with intercept blocking buffer (#92760001, LI-COR Biosciences, Lincoln, NE, USA) at RT for 1 h. It was subsequently incubated at 4 °C overnight with a gentle shaking shaker with the following monoclonal primary antibodies: mouse anti-TGFβ1 (sc-130348, Santa Cruz Biotechnology, Dallas, TX, USA, 1:100), rabbit anti-IRF3 (D83B9, Cell Signaling Technology, Danvers, MA, USA; 1:1000), rabbit anti-STAT-3 (#9132, Cell Signaling Technology; 1:1000), mouse anti-PPAR-γ (SC-7273, Santa Cruz Biotechnology, Dallas, TX, USA; 1:200), mouse anti-STAT-5 (#SC-74442, Santa Cruz Biotechnology; 1:200,) and mouse anti-β-actin (#926-42212, LI-COR Biosciences, Lincoln, NE, USA; 1:1000). After washing, membranes were incubated with IRDye^®^ 800CW-labeled goat anti-mouse IgG (#926-32210, LI-COR Biosciences; 1:5000) and IRDye^®^ 800CW-labeled goat anti-rabbit (#926-32211, LI-COR Biosciences; 1:5000) secondary antibodies at RT for 1 h. Signal intensity from each immunoblot was measured using the LI-COR Biosciences Odyssey Sa Infrared Imaging system. We repeated the experiment three times and presented the data as mean ± SEM for each experimental condition.

### 2.8. Histology

The epididymal AT samples were thoroughly washed in phosphate-buffered saline (PBS) and cut longitudinally to closely similar sizes for each experimental group. This tissue was fixed in 4% paraformaldehyde for 24 h and embedded in paraffin. Fixed tissues were sectioned at 4 µm and stained with hematoxylin and eosin (H&E) for the examination of adipocytes by light microscopy. AT sections were examined for assessment of inflammation, adipocyte size, and infiltration of immune cells.

### 2.9. Statistics

All statistical analyses were performed using the stat view II statistical program (Abacus Concepts, Inc., Berkeley, CA, USA) for Macintosh computers. The data were presented as the mean of the standard error (± SEM) or *n* (%) for flow cytometry. The difference in continuous variables within the two groups (HFD and ND) was determined by the student’s *t*-test. We also used *t*-tests within the context of two groups and ANOVA to show whether any differences in mean cytokine levels and flow cytometry data were statistically significant. The results were analyzed using the unpaired Student’s *t*-test, and *p* values less than 0.05 were considered to be statistically significant (* *p* < 0.05).

## 3. Results

### 3.1. Changes in Body Weight and Metabolic Parameters after HFD Induction

During obesity, the infiltration of immune cells into AT is associated with low-grade chronic inflammation, thereby altering metabolic diseases associated with obesity. Within AT, crosstalk between immune cells and adipocytes is key for sustaining AT inflammation and can boost the effects of obesity. We determined the changes in body weight and other metabolic parameters in mice fed with a normal diet (ND) and those fed with a high-fat diet (HFD). The body weight of mice that received HFD increased significantly (*p* < 0.05) compared to mice fed with ND (Figure 1A). We also observed a significant difference (*p* < 0.05) in fat mass and water content in mice fed HFD as compared to those fed with ND (Figure 1B). Furthermore, a slight increase in lean mass was observed in mice fed HFD as compared to those fed ND. HFD-fed mice exhibited a significant increase (*p* < 0.05) in blood glucose and insulin levels, as compared to mice fed ND (Figure 1C,D). We also observed a corresponding increase in the size of adipocytes in mice fed HFD as compared to those fed ND (Figure 1E,F). Taken altogether, these results indicated that HFD alters the body weight and other metabolic markers associated with metabolic syndrome in AT, thereby augmenting obesity.

### 3.2. HFD Induces M1 Macrophages, CD103^+^ Dendritic Cells, and Monocytic Myeloid-Derived Suppressor Cells (MDSCs) in AT

Macrophages are essential components of the innate immune system and are important for host defense against inflammation. Macrophages of the proinflammatory M1 phenotype express CD11c and promote a Th1 response to mediate chronic AT inflammation. To determine whether HFD induces M1 macrophage phenotype and function in mice, we used flow cytometry to analyze the changes in macrophage phenotypes in the spleen, MLNs, and AT-derived cells of mice fed ND or HFD. The frequency and percentage of pro-inflammatory M1 macrophages (CD11b^+^CD11c^+^) increased significantly (*p* < 0.01) in the AT of HFD-fed mice as compared to those fed ND (Figure 2A,D). We did not observe any changes in the macrophage phenotype in the spleen or MLNs from either group. These results suggest that HFD induces macrophages to switch to a pro-inflammatory M1 phenotype, presumably responsible for sustained low-grade chronic inflammation in the AT.

DCs play a key role in balancing tolerance and immunity and are involved in the recruitment and activation of macrophages at the site of inflammation [29]. DCs are elevated in obesity, where they promote macrophage infiltration in AT and alter the systemic metabolic response to HFD [30]. To determine whether HFD induces DCs in mice, we used flow cytometry to analyze CD103^+^ DCs in the AT, spleen, and MLNs of mice fed ND or HFD. The frequency and percentage of CD11b^+^CD103^+^ DCs were increased significantly (*p* < 0.01) in the AT of mice fed HFD as compared to ND (Figure 2C,G). In contrast, a slight decrease in DCs in the spleen (Figure 2A,C) and MLNs (Figure 2A,C,G) was observed in the HFD as compared to ND groups. The results suggest that HFD induces infiltration of DCs into AT, which is likely to promote macrophage infiltration or Th17 response to sustained AT inflammation.

MDSCs are a heterogeneous population of immature myeloid cells that expand under pathological conditions and are crucial for the prevention of chronic inflammation and immune response [31,32]. Monocytic (m-MDSC) and granulocytic (g-MDSC) subsets of MDSCs are elevated both in mice fed on HFD and in obese (ob/ob) mice that are genetically deficient for leptin expression [33]. The mechanisms by which MDSCs alter AT inflammation remain unclear. Therefore, we used flow cytometry to determine whether we would observe any changes in frequency and percentage of AT MDSCs in HFD-fed mice. We did not notice any difference in the frequency of MDSCs in the spleen and MLNs of HFD-fed mice as compared to ND-fed mice (Figure 2B,G). However, we observed a significant increase (*p* < 0.01) in m-MDSCs in the AT of mice that were fed HFD as compared to the ND-fed controls (Figure 2B,G). The induced m-MDSCs in AT were either not immunosuppressive or were likely to promote inflammation. However, we would need to examine the suppressive function of purified m-MDSCs and g-MDSCs from AT in HFD- or ND-fed mice to reach a substantial conclusion in this regard.

### 3.3. Th17/Tregs Dysregulation in AT Correlates with Inflammation

During obesity, excess lipids and glucose can directly impact Th17 and Treg cell activation and differentiation through nutrient sensor activity. Th17 cells are associated with obesity-induced inflammation [11], while the number of regulatory T cells (Tregs) is reduced under obese conditions [14]. It is unclear whether this Th17/Treg imbalance influences obesity and metabolism or is simply enhanced by obesity-associated inflammatory changes. We compared the frequency of Th17/Tregs found in the AT, spleen, and MLN of mice fed HFD or ND. The AT, spleen, and MLN of HFD-fed mice contained significantly (*p* < 0.01) fewer Tregs than did those of mice fed ND (Figure 3A,C). In contrast, we observed a significant increase (*p* < 0.01) in Th17 cells in these tissues from mice fed HFD (Figure 3B,D). These findings suggest that the balance of Th17/Tregs in AT plays opposite roles in normal and obese conditions. In the normal healthy condition, numbers of Th17/Tregs are kept in balance, while their relative proportions are altered during pathological conditions such as obesity, where a marked increase in Th17 and reduction in Tregs serves to induce and sustain chronic inflammation.

### 3.4. HFD-Induced Th17 Mediates Th1 and M1 Macrophage Response in AT

Interleukin-17 (IL-17) plays an important role in promoting obesity-associated AT inflammation by inducing the expression of various chemokines that recruit leukocyte tissue infiltration and exacerbate AT inflammation. However, we find it puzzling that a few studies show that IL-17 can delay the development of obesity [34]. Therefore, we enumerated the changes in the immune responses of Th17 cells from systemic or AT environments after feeding mice on HFD or ND. In AT, we observed the accumulation of a higher percentage of Th17 cells that also express IFN-γ in HFD-fed mice compared to mice fed ND (Figure 4A,B). However, we did not observe any changes in Th17 cells in the spleens or MLNs of mice fed either diet.

Further, we also performed functional studies to determine whether HFD induction polarizes M1 macrophage response in AT. For this, we enumerated the changes in M1 macrophage (CD11b+F4/80+CD11c) frequency after HFD feeding. We observed an increase in the M1 macrophage in AT of mice that received HFD as compared to ND (Figure 4C,D). Further, we also observed that HFD-derived M1 macrophages mainly augmented TNF-α production as compared to ND-fed mice (Figure 4C,D). Thus, results confirm that HFD induces mainly M1 macrophages to induce inflammation in AT. This result, combined with our previous observation that HFD induces a Th1 and M1 macrophage response in AT, further suggests that HFD also induces a Th17 response that might enhance the production of IFN-γ to sustain chronic inflammation in AT.

### 3.5. HFD Differentially Mediates Th1/Th17 Cytokine and Chemokine Response

Adipose resident immune cells regulate inflammation through cytokine and chemokine secretion, which share some common features [35]. We used a Bio-Plex ELISA assay to measure the levels of common chemokines and cytokines in serum from mice fed HFD or ND. We observed that HFD-fed mice exhibited decreased levels of IL-5, IL-6, and G-CSF compared to mice that were fed ND (Figure 5A,D). In contrast, obese mice exhibited enhanced levels of inflammatory cytokines and chemokines RANTES, IL-17, IL-13, IP-10, MIP-1β, IL-10, and TNF-α (Figure 5B,C,E). Taken together, these data suggest that HFD plays a role in inducing chronic inflammation, by increasing the Th1 response in immune cells and adipocytes in AT, which might be at least partially responsible for sustained AT inflammation.

### 3.6. HFD Induced Differential Signaling Pathways to Sustain Chronic AT Inflammation

Next, we sought to determine which signaling pathways in adipose-derived cells might be induced by HFD to maintain chronic inflammation in AT. The interferon regulatory factor 3 (IRF3) plays an important role in obesity and the regulation of insulin sensitivity. PPAR-γ is considered the master regulator of adipogenesis and is involved in adipocyte differentiation and glucose metabolism. IL-6 activates the signal transducer and activator of transcription 3 (STAT3) to differentiate Th17 cells from naïve CD4^+^ precursor cells. To test the relative effect of HFD on these signaling pathways, we performed RT-qPCR and immunoblot analysis on these molecules in the AT from mice fed HFD or ND. The AT of HFD-fed mice exhibited increased expression of IL-6, KLF4, and STAT3 transcripts (*p* < 0.01), as compared to that of ND-fed mice (Figure 6A). Further, levels of IRF3 and TGFβ1 protein increased in the AT of HFD-fed mice as compared to that of ND-fed mice (Figure 6B). We also observed a significant decrease in PPAR-γ (*p* < 0.01) expression in the AT from the HFD group as compared to that from the ND group (Figure 6B). In contrast, we also noticed a slight decrease in STAT3 expression in AT of the HFD-fed mice. Taken together, these data suggest that HFD induces the IRF3, TGFβ1, and STAT3 signaling pathways in AT that in part might suppress the expression of PPAR-γ to maintain low-grade chronic inflammation in AT during HFD-induced obesity.

## 4. Discussion

Obesity is a major worldwide problem that affects more than 2 billion adults worldwide. Obesity represents a risk factor for many diseases ranging from insulin resistance, autoimmune diseases, and T2D to cardiovascular and metabolic syndrome. Obesity is associated with chronic low-grade inflammation in AT without any effective therapy. A key to understanding this process is the discovery of the factors and parameters that regulate immune cell infiltration in AT and their roles in sustaining chronic inflammation under obese conditions. Therefore, in this study, we investigated the relationship between AT and immune cells that alter the balance of Th17/Tregs during HFD-induced obesity to mediate the function of macrophages and other cells in AT to sustain chronic inflammation. We found that increased infiltration of Th17 and DCs led to an accumulation of M1 macrophages and observed an increase in the proteins involved in IRF3-STAT3 signaling in the AT. Further, a decrease in PPAR-γ expression in obese AT suggests that these pathways serve as negative regulators of Th17/Treg function. Taken together, our results suggest that HFD induces Th17, CD103^+^dendritic cells, and pro-inflammatory cytokines and reduces Tregs and PPAR-γ that in part polarize M1 macrophages, which might be responsible for sustaining AT inflammation. These data support a role in part for Th17/Treg dysregulation, which facilitates recruitment of DCs and macrophages and induces a phenotypic switch in macrophages to M1, which mediate AT inflammation through the IRF3-STAT3 and PPAR-γ signaling pathways during obesity.

IL-17A is an important regulator of glucose homeostasis, adipogenesis, and obesity [34,36] that also plays a critical role in the crosstalk between obesity and Th17-associated immune response in AT. Levels of circulating Th17 cells are significantly higher in obese groups than in normal lean groups and positively correlate with BMI [10,11]. Infiltration of AT by Th17 cells increased in obese individuals and mice with diet-induced obesity [12]. Our results showing a significant increase in Th17 cells in AT and systemic levels in mice fed HFD lend support to the suggestion that Th17 cells mediate the inflammatory response in AT through major effector cytokines IL-6 and IL-17. As obesity progresses, the numbers of Tregs decrease in obese VAT [14,15,37]. Similarly, the decrease in Tregs and resulting Th17/Treg imbalance are inversely correlated with BMI [16,17,18]. In this study, we observed a decrease in the frequency of Tregs in mice fed HFD relative to ND-fed mice, which substantiates the previous findings published by others. The reduction in Tregs failed to maintain immune homeostasis and to defend against obesity-associated metabolic inflammation in AT. Thus, these results strongly suggest that the Th17/Treg imbalance plays a key role in the progression and maintenance of obesity-dependent inflammation in AT. However, a direct functional study in which purified Treg/Th17 cells from the AT of HFD-fed mice are adoptively transferred to obese (ob/ob) mice might lend more conclusive support to this conclusion.

Macrophages and DCs are both antigen-presenting cells (APCs) and key participants in the innate and adaptive immune responses during obesity and related metabolic syndrome. It should be clear by now that the numbers of M1 macrophages in AT increase and correlate with AT inflammation and IR [23]. A recent study of diet-induced obesity detected recruitment into AT of a specific subset of macrophages that express low levels of CD11c [26]. Furthermore, specific subsets of DCs that express macrophage markers in obese mice induced differentiation of Th17 cells [27]. Similarly, in humans, the presence of DCs in AT correlates with BMI and elevation of Th17 cells. In the present study, we observed an increase in both M1 macrophage frequency as well as in TNF-α secretion by these cells and CD103^+^ DCs in the AT of HFD-fed mice, as compared to those fed ND. It has been shown that TGFβR signaling controls CD103^+^DCs in the intestine [38]. In this study, we also observed an increase in TGFβ1 expression in HFD-fed mice, suggesting similar pathways in which TGFβ1 induces CD103^+^DCs in AT during obesity. This observation supports and corroborates the previous findings by others on M1 macrophage function during obesity and the correlation of AT DCs with obesity-associated IR. This also suggests the possibility that CD103^+^ DCs promote the infiltration of M1 macrophages during HFD-induced obesity. Thus, the results of this study strongly suggest the existence of a crosstalk between DCs and macrophages in obese AT that might propagate and sustain local AT inflammation.

A higher level of pro-inflammatory cytokines is observed in both genetically obese (ob/ob) and HFD-fed mice. In obese or overweight adult individuals, excess fat secretes more inflammatory cytokines into the circulation [39,40]. After activation of macrophages, levels of TNF-α, IL-6, and RANTES elevated during obesity maintain homeostatic control of AT mass [41,42]. The cytokine granulocyte-colony stimulating factor (G-CSF) is known for its anti-obesity effects, while IL-10 is anti-inflammatory and is present at lower circulating levels in obese subjects. In this study, we observed an increase in systemic concentrations of IL-13, IL-17, IP-10, MIP1-β, RANTES, and TNF-α in the serum of HFD-fed mice relative to the levels observed in mice fed ND. These data corroborate previous observations that during obesity, Th17/M1 macrophages induce a Th1 response in the AT and increase the production of cytokines and chemokines [43,44]. Our observed decrease in cytokines IL-5 and G-GCF in HFD-fed mice supports the previous findings of the anti-obesity effects of these cytokines [45]. Taken altogether, our data suggest that HFD leads to the expansion of Th17 and M1 macrophages that secrete mainly Th1-biased inflammatory cytokines and chemokines and also alters the levels of IL-10, IL-13, IL-5, and G-CSF. The increased levels of IL-10 and IL-13 and decreased levels of IL-5 might be involved in maintaining immune tolerance during the initial stage of obesity, but later, might indirectly support M1 macrophage function to maintain chronic inflammation, as described previously [46].

STAT3 initiates signaling pathways through IL-6 and Th17 differentiation [47,48]. PPAR-γ is involved in adipocyte differentiation, glucose metabolism, and a deficiency of macrophages, resulting in increased obesity-induced AT inflammation [49]. PPAR-γ also reverses macrophage infiltration and subsequently reduces the expression of pro-inflammatory genes [35]. The novel regulator of macrophage polarization Kruppel-like factor 4 (KLF4) also controls adipogenesis [50], while IRF3 plays an important role in the regulation of insulin sensitivity, inflammation, and obesity [51]. The mitochondrial stress-activated pathway functions as a custodian signal that suppresses AT thermogenesis and IRF3 pathways are critically involved in metabolic stress-induced inflammation [52]. We observed increased transcript expression of IL-6, KLF-4, and STAT3 and protein expression of TGFβ and IRF-3 in AT from HFD-fed mice, as compared to mice fed ND. Further, we observed significantly decreased levels of PPAR-γ and a slight decrease in STAT3 expression in the HFD-fed as compared to the ND-fed group. Taken together, our data support the hypothesis that HFD induces differentiation of Th17 cells through IL-6, TGFβ1, and STAT3 to induce adipogenesis, which might downregulate PPAR-γ to recruit more M1 macrophages that sustain chronic inflammation. Once chronic inflammation is initiated, the IRF3-KLF4 pathways are induced and play a role to maintain AT inflammation. This study was performed using total AT that included both adipocyte and infiltrating immune cells, which might be a reason for some of the discrepancies in STAT3 levels. Therefore, further studies are required using purified AT M1 macrophages and Th17/Treg cells to determine the role of these signaling pathways in AT.

In summary, our results suggest that HFD-induced obesity in mice induces an influx of CD103^+^DCs to the AT that alters the Th17/Treg balance, leading to recruitment and activation of macrophages, whose subsequent switch to the M1 phenotype and production serves to sustain chronic inflammation. This study supports a key role of Th17/Treg dysregulation that induces AT inflammation through the production of Th1 cytokines and chemokines that signal via the TGFβ1-IRF3-STAT3 and PPAR-γ pathways. However, more functional studies, including the use of adoptive transfer of purified Th17/Tregs and M1 macrophages, will be required before we can reach any firm conclusions on their precise role in AT inflammation.

## Figures and Tables

**Figure 1 cells-11-01327-f001:**
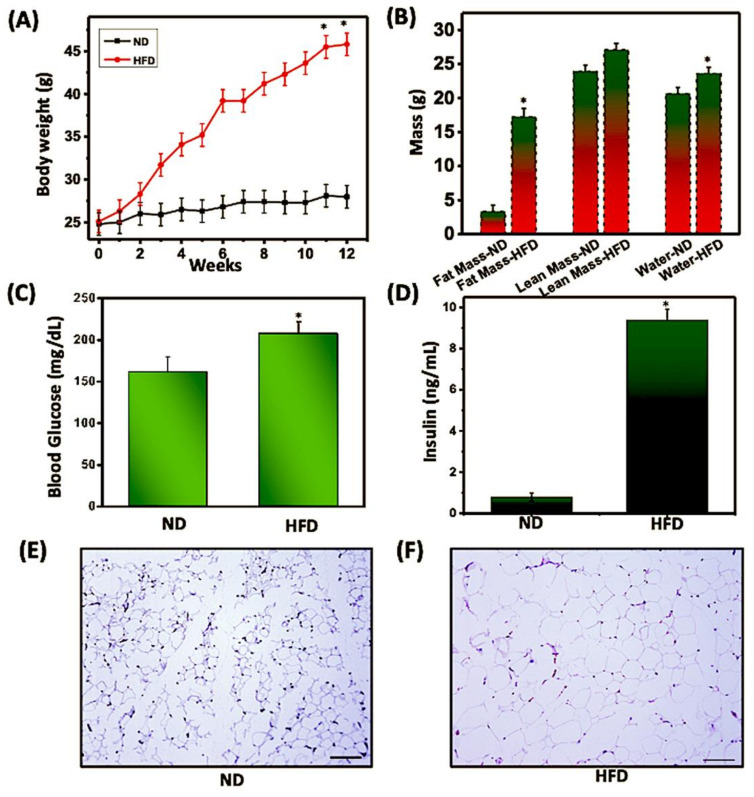
**HFD altered the metabolic profiles and body composition.** Mice were fed a normal diet (ND) or high-fat diet (HFD) for 12 weeks before sacrifice. (**A**) Mice fed ND or HFD were weighed weekly, and changes in body weight were recorded. (**B**) Echo MRI analysis of ND and HFD mice to measure whole-body composition, including fat mass (grams), lean body mass (grams), and water content (grams). Without fasting, plasma (**C**) glucose and (**D**) insulin levels were measured at the experimental endpoint. Representative histopathology of AT with H&E staining from mice fed ND (**E**) or HFD (**F**). Sections were examined microscopically at a magnification of 10X by light microscopy. Scale bars on the (right), 100 μm. The statistical significance between values of each group was assessed by an unpaired Student’s *t*-test. * *p* < 0.05 indicates statistically significant differences in body weight, body fat, fasting blood glucose, and plasma insulin between mice fed ND or HFD. Values are mean ± SEM; total *n* = 12 (six mice per group). Data are representative of the mean of three independent experiments.

**Figure 2 cells-11-01327-f002:**
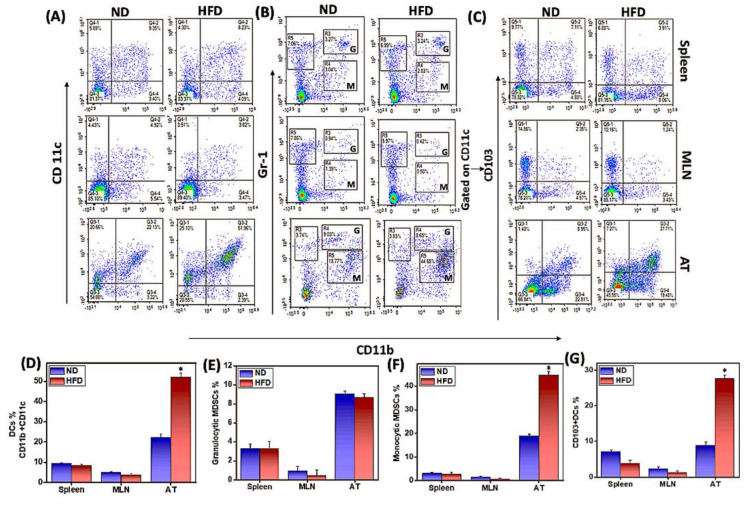
**HFD induces M1 macrophages (CD11b^+^CD11c^+^), monocytic MDSCs (CD11b^+^ LY6c^High^ LY6g^Low^), and CD11b^+^CD103^+^ dendritic cells (DCs) in adipose tissue (AT).** Mice were fed HFD or ND for 12 weeks before sacrifice. Spleens, mesenteric lymph nodes (MLNs), and epididymal adipose tissue (AT) were harvested from each group of mice and single-cell suspensions were prepared from each organ using a tissue stomacher and analyzed by flow cytometry. Changes in the frequency of (**A**) M1 macrophages, (**B**) granulocytic (G) or monocytic (M) myeloid-derived suppressor cells (MDSCs), and (**C**) CD103^+^ dendritic cells from HFD-fed mice flow cytometry data were expressed as the mean percentage of cells/mouse ± SEM in a bar graph (**A**–**D**, **B**–**E**, **M**–**F**, **C**–**G**). Data shown are from a representative experiment; three independent experiments involving six mice/groups yielded similar results. Asterisks (*) indicate statistically significant differences (*p* < 0.01) between ND and HFD groups.

**Figure 3 cells-11-01327-f003:**
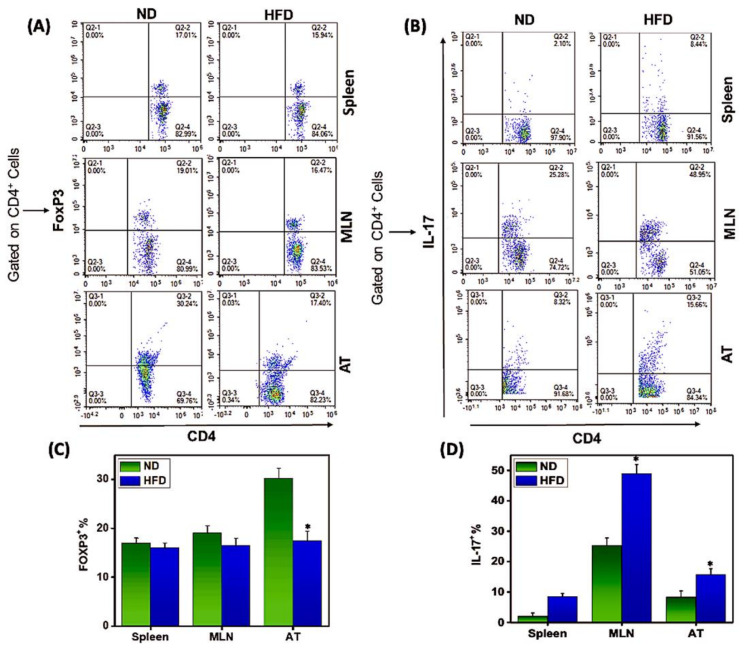
**HFD-induced obesity differentially mediates the expression of Th17/Tregs in AT.** Cells from spleens, MLNs, and AT were isolated from mice fed ND or HFD as described in the legend in Figure 2 and analyzed by flow cytometry gated on CD4^+^ T cells and antibodies specific for CD4, Foxp3, and Th17. (**A**,**B**) Changes in AT in the frequency of (**A**) CD4^+^FoxP3 ^+^ Tregs or (**B**) CD4^+^ Th17^+^ Th17 cells from mice fed HFD or ND. (**C**,**D**) Percentage change in (**C**) Foxp3^+^ and (**D**) Th17^+^ in AT from mice fed HFD or ND. Relative values are expressed as the mean percentage of cells/mice ± SEM. Data shown are from a representative experiment; three independent experiments involving six mice/groups yielded similar results. Asterisks (*) indicate statistically significant differences (*p* < 0.01) between ND and HFD groups.

**Figure 4 cells-11-01327-f004:**
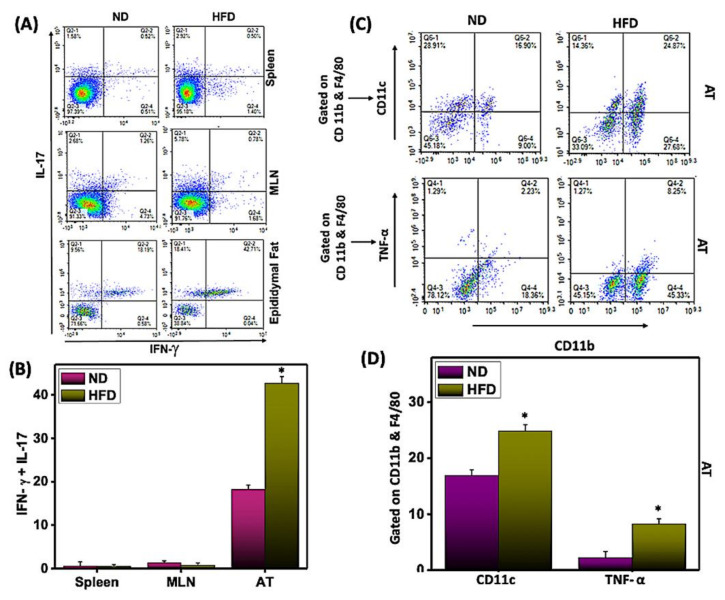
**HFD induces IFN-γ^+^ Th17^+^ cells in AT.** AT-derived cells were isolated from the two groups of mice as described in Figure 2 and analyzed by flow cytometry with antibodies specific for Th17 and IFN-γ. Changes in the frequency (**A**) and percentage of Th17^+^IFN-γ^+^ cells (**B**) in the systemic and AT environments were expressed as the mean percentage of cells/mice ± SEM. Further, changes in the frequency and percentage of M1 macrophage (CD11c and TNF-α gated on CD11b+F4/80) cells (**C**,**D**) in the AT were expressed as the mean percentage of cells/mice ± SEM. Histograms are shown from a representative experiment. The percentage change is shown by combining the mean of the three independent experiments involving six mice/groups that yielded similar results. Asterisks (*) indicate statistically significant differences (*p* < 0.01) between ND and HFD groups.

**Figure 5 cells-11-01327-f005:**
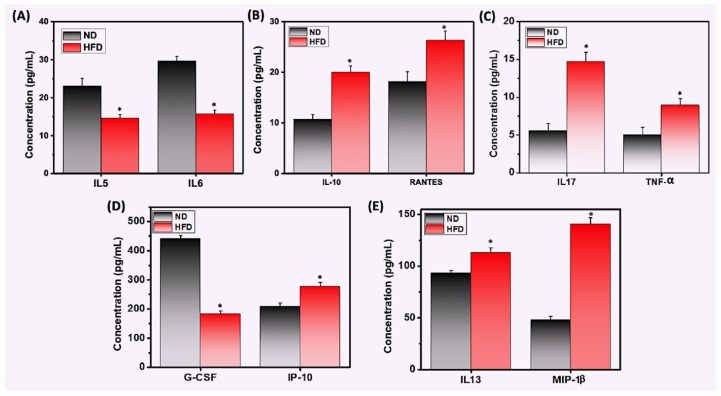
**HFD differentially induces the expression of cytokines and chemokines in the serum of mice fed HFD or ND.** Mice were fed HFD or ND for 12 weeks before sacrifice, and whole blood was collected from each group. Serum samples were prepared and analyzed by ELISA assay using antibodies specific for cytokines and chemokines G-CSF, IL-5, IL-6, IL-10, IL-13, IL-17, IP-10, MIP-1β, RANTES, and TNF-α. (**A**) HFD suppressed systemic expression of IL-5 and IL-6 proteins, (**B**,**C**,**E**) increased levels of systemic IL-10, IL-13, IL-17, IP-10, MIP-1β, RANTES, and TNF-α, and (**D**) suppressed levels of G-CSF, all relative to levels observed in mice fed ND. Data represent the levels of inflammatory cytokines and chemokines ± SEM from three independent experiments. Asterisks (*) indicate statistically significant differences (*p* < 0.01) between ND and HFD groups.

**Figure 6 cells-11-01327-f006:**
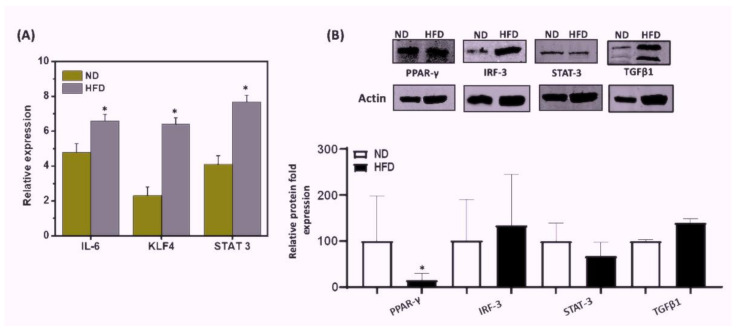
**Effects of HFD or ND feeding on mRNA expression and protein levels in mouse AT.** Mice were fed HFD or ND for 12 weeks before sacrifice, and AT was collected from each group. (**A**) RT-qPCR analysis of IL-6, KLF-4, and STAT3 transcripts in AT was obtained from mice fed with ND or HFD. Total RNA was isolated from AT from each group of mice, pooled, quantitated, reverse-transcribed into cDNA, and analyzed by qPCR with primers specific for IL-6, KLF-4, and STAT3. HFD mice expressed higher levels of IL-6, KLF-4, and STAT3 transcripts than did mice fed ND. (**B**) Immunoblot analysis of PPAR-γ, TGFβ1, IRF3, and STAT3 protein expression levels in mice fed for 12 weeks with ND. AT from mice fed HFD or ND was lysed to obtain whole cell lysates and relative levels of TGFβ1, IRF3, PPAR-γ, and STAT3; proteins were determined by immunoblot analysis using specific monoclonal antibodies. HFD mice exhibited higher levels of protein expression of TGFβ1 and IRF3 than did mice fed ND. In contrast, protein expression levels of PPAR-γ and STAT3 were decreased in HFD-mice relative to the ND control. Vertical bars represent mean SEM, and significant differences between the groups are shown with (*) (*p* < 0.01) based on one-way ANOVA.

## Data Availability

The raw data, including flow cytometry data, supporting the conclusions of this article will be made available to the public by the authors, without any reservation.

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
