# Peer review of "High-Fat Diet-Induced Dysregulation of Immune Cells Correlates with Macrophage Phenotypes and Chronic Inflammation in Adipose Tissue"

_cells, 2022, doi:10.3390/cells11081327_

Round 1

Reviewer 1 Report

General:

In this study the authors use a high fat diet (HFD) fed mouse model to explore their hypothesis that HFD induces inflammation in adipose tissue by dysregulating Th17 and Tregs repertoire to polarize macrophages towards an M1 phenotype promoting chronic inflammation.

In general the manuscript is well-organised and written. The hypothesis is interesting and the data in general seems thorough, however in its current form I would suggest this investigation is generally descriptive of a HFD induced phenotype with associations highlighted – rather than as the authors suggest (on multiple occasions) that their data demonstrates a mechanism.

However, the novel data presented will be useful to the scientific community if formatting errors are addressed and the text modified to reflect the data’s more descriptive nature; in the introduction for example:

“our results suggest that HFD induces Th17 cells, CD103+DCs, inflammatory cytokines and reduces Tregs in part through cGAS-IRF3, STAT3, KLF4, and PPAR-γ mediated pathways, which polarize macrophages towards M1 phenotypes to maintain 101 low-grade AT inflammation.”

  • While I agree with the first half of this statement, the authors do not present any mechanistic studies (e.g inhibitor experiments) which link those intracellular pathways to Treg numbers or macrophage polarisation

Methods and materials:

In general, comprehensively written. The manuscript could be significantly improved by including females in their main study as sex-specific differences are topical and important to include in preclinical studies. The authors state female mice are used in ongoing functional studies but it is unclear in the results which experiments may include female data too. However this does not detract from the novelty and accuracy of the male data presented.

Results:

For all figures:

The formatting needs to be improved ensuring consistency between each figure. For example text size of figure axes and spelling (eg Fig1D).

Asterix’s (*) are not aligned over specific data point (Fig.1A) and are presented above the normal diet control model (Figs 1-3) or the experimental high fat diet (HFD) model (Fig4-6) – therefore it is challenging to be sure which data is being compared to which group.

Language is too vague when describing the data and needs to be re-written,  e.g “did not change much” (line 319).

Specific Figs:

Fig1A body mass data for week 0?

Flow cytometry panels could be expanded in size and resolution.

Results text (line 271) states that adipocytes are larger in HFD model, however Fig1E&F in its current form does not reflect this to my eyes, and there are no scale bars or magnification information or graphical analysis (which could show number of cells on how many sections investigated etc).

Fig 6 needs to be expanded to show all the findings, or the negative data should be included in a supplementary figure so we can see consistency between the gene expression data and protein data for all the genes investigated. The western blot for C-GAS is not particularly convincing when taking into account the control actin blot.

Discussion:  

I am not convinced the authors investigated the “underlying mechanisms” as stated in line 461 and I would suggest changing the text to describing the novel associations of the study. The authors should also mention their thoughts on how a high fat diet (or components of) may cause the observations they are describing. The discussion section could also be significantly improved if the authors could include a concise summary figure of their findings including their speculation on how the intracellular pathways may produce the phenotype observed (cytokines and cell recruitment/ polarization).

Author Response

Reply to Reviewers 1 Comments: All changes are underlined in the Text.

In this study the authors use a high-fat diet (HFD) fed mouse model to explore their hypothesis that HFD induces inflammation in adipose tissue by dysregulating Th17 and Tregs repertoire to polarize macrophages towards an M1 phenotype promoting chronic inflammation. In general, the manuscript is well-organized and written. The hypothesis is interesting and the data, in general, seems thorough, however, in its current form, I would suggest this investigation is generally descriptive of an HFD induced phenotype with associations highlighted – rather than as the authors suggest (on multiple occasions) that their data demonstrates a mechanism. However, the novel data presented will be useful to the scientific community if formatting errors are addressed and the text is modified to reflect the data’s more descriptive nature; in the introduction for example:

  1. Our results suggest that HFD induces Th17 cells, CD103+DCs, and inflammatory cytokines and reduces Tregs in part through cGAS-IRF3, STAT3, KLF4, and PPAR-γ mediated pathways, which polarize macrophages towards M1 phenotypes to maintain low-grade AT inflammation. While I agree with the first half of this statement, the authors do not present any mechanistic studies (e.g., inhibitor experiments) which link those intracellular pathways to Treg numbers or macrophage polarization.

Response: We were very grateful for the comments and suggestions by the reviewers. We have carefully made corrections and modifications to improve the quality of the manuscript based on the reviewer's comments. We modified the second-half statement by removing the pathways and only correlating those pathways with changes in immune cells during adipose tissue inflammation.  

Methods and materials:

  1. In general, comprehensively written. The manuscript could be significantly improved by including females in their main study as sex-specific differences are topical and important to include in preclinical studies. The authors state female mice are used in ongoing functional studies but it is unclear in the results which experiments may include female data too. However, this does not detract from the novelty and accuracy of the male data presented.

Response: We were very grateful for the comments and suggestions of the reviewers. We have used male mice only for this study, as well as other previous studies.  In our earlier study, we used 8-week old mice as our starting points. It has been shown in several studies that male mice had a greater propensity of gaining bodyweight than female mice. Further, it has been also shown that obese male mice produce more disease-promoting immune cells than females (Journal of Biological Chemistry, 2015; 290 (21):13250.

Results:

For all figures:

The formatting needs to be improved to ensure consistency between each figure. For example, text size of figure axes and spelling (eg Fig1D).

Asterix’s (*) are not aligned over specific data points (Fig.1A) and are presented above the normal diet control model (Figs 1-3) or the experimental high-fat diet (HFD) model (Fig4-6) – therefore it is challenging to be sure which data is being compared to which group.

Language is too vague when describing the data and needs to be re-written,  e.g “did not change much” (line 319).

Specific Figs: Fig1A body mass data for week 0?

Flow cytometry panels could be expanded in size and resolution.

Results text (line 271) states that adipocytes are larger in the HFD model, however Fig1E&F in its current form does not reflect this to my eyes, and there are no scale bars or magnification information or graphical analysis (which could show several cells on how many sections investigated, etc).

Fig 6 needs to be expanded to show all the findings, or the negative data should be included in a supplementary figure so we can see consistency between the gene expression data and protein data for all the genes investigated. The western blot for C-GAS is not particularly convincing when taking into account the control actin blot.

Response: We were very grateful for the comments and sorry for these oversights. We have modified the revised manuscript based on all suggestions. We corrected all the points in the Figures and results in sections.

Discussion:  

I have not been convinced the authors investigated the “underlying mechanisms” as stated in line 461 and I would suggest changing the text to describe the novel associations of the study. The authors should also mention their thoughts on how a high-fat diet (or components) may cause the observations they are describing. The discussion section could also be significantly improved if the authors could include a concise summary figure of their findings including their speculation on how the intracellular pathways may produce the phenotype observed (cytokines and cell recruitment/ polarization).

Response: We were very grateful for the comments. We have modified the revised manuscript based on suggestions. We corrected all the points and included a summary figure that described the pathways and observed phenotypes.

Reviewer 2 Report

The manuscript submitted by Kiran et al, titled “High-fat diet-induced dysregulation of immune cells correlates with macrophage phenotypes and chronic inflammation in adipose tissue” investigated the underlying mechanisms of immune dysregulation in high fat diet (HFD) obesity in in-bred mice. The manuscript does quite comprehensive characterization of the immuno-phenotype in the mice fed HFD. Their data demonstrates that there is a profound decrease in Treg levels in mesenteric lymph nodes (MLN), spleen and epididymal adipose tissue (AT). This is accompanied with an increase in the levels of total and IFNg producing TH17 cells as well as increase in the proportion of M1 macrophages and CD103+ DC in the AT. The authors also show an increase in the expression of several Th1 response associated cytokines and chemokines in the circulation. The authors finally demonstrates that the levels of IL-6, KLF-4 and STAT3, which are important regulators of the TH17 immune response are upregulated in the AT along with a decrease in PPARg expression. The data overall supports the conclusion of the manuscript that HFD-induced metabolic changes alter the immunophenotype to a more pro-inflammatory one thus contributing to chronic inflammation. The article furthers the reader’s understanding of immune dysregulation by HFD. The reviewer has the following comments to further improve the manuscript:

  1. The Th17 response is induced by TGFb along with IL-6. Have the authors measured the levels of TGFb in the AT as a likely contributor ? In Fig 3C, the levels of CD103+ DC are shown by flow plots depicting CD103 and CD11b. Were these cells gated on the CD11c+ cells? Furthermore, the development of the mucosal CD11b+CD103+ DC was shown to be regulated by TGFb signaling (PMID 28931816). Thus knowing the levels of TGFb in the AT may explain the shift in the immunophenotype.
  2. Are the blood glucose measurements shown in Fig 1 performed at fasting or are they fed-blood glucose levels? Please clarify.
  3. The statement in lines 522 that “ increased levels of IL-10 and IL-13…..would maintain chronic inflammation” is not substantiated. IL-10 is an immunoregulatory cytokine and its increase may be an effect to counter chronic inflammation rather than being a cause. The reviewer suggests modification of this sentence.
  4. cGAS/STING are nucleic acid sensors and not expected to increase in HFD studies. Since cGAS higher and STING is lower in AT in panel 6D, it distracts from the main conclusion of the study and the reviewer suggests removing these.
  5. The acronym MS for metabolic syndrome is confusing, because this is commonly used for multiple sclerosis.
  6. In the introduction section, the authors mention TH17 and Tregs repertoire in line 95. This should be changed to subsets, because the repertoire for T-cells is used for T-cell receptors diversity and is misleading.
  7. Correct the spelling of Insulin in Y-axis label in Fig 1D.
  8. In line 365, the authors state that “a few studies show IL-17 can delay the development of obesity”. This needs a citation.

Author Response

Reply to Reviewers 2 Comments: All changes are underlined in the Text.

The manuscript submitted by Kiran et al, titled “High-fat diet-induced dysregulation of immune cells correlates with macrophage phenotypes and chronic inflammation in adipose tissue” investigated the underlying mechanisms of immune dysregulation in high-fat diet (HFD) obesity in inbred mice. The manuscript does a quite comprehensive characterization of the immuno-phenotype in the mice fed HFD. Their data demonstrate that there is a profound decrease in Treg levels in mesenteric lymph nodes (MLN), spleen, and epididymal adipose tissue (AT). This is accompanied by an increase in the levels of total and IFNg producing TH17 cells as well as an increase in the proportion of M1 macrophages and CD103+ DC in the AT. The authors also show an increase in the expression of several Th1 response-associated cytokines and chemokines in the circulation. The authors finally demonstrate that the levels of IL-6, KLF-4, and STAT3, which are important regulators of the TH17 immune response are upregulated in the AT along with a decrease in PPARg expression. The data overall supports the conclusion of the manuscript that HFD-induced metabolic changes alter the immunophenotype to a more pro-inflammatory one thus contributing to chronic inflammation. The article furthers the reader’s understanding of immune dysregulation by HFD. The reviewer has the following comments to further improve the manuscript:

  1. The Th17 response is induced by TGFb along with IL-6. Have the authors measured the levels of TGFb in the AT as a likely contributor ? In Fig 3C, the levels of CD103+ DC are shown by flow plots depicting CD103 and CD11b. Were these cells gated on the CD11c+ cells? Furthermore, the development of the mucosal CD11b+CD103+ DC was shown to be regulated by TGFb signaling (PMID 28931816). Thus, knowing the levels of TGFb in the AT may explain the shift in the immunophenotype.

Response: We were very grateful for the comments and suggestions of the reviewers. We have carefully made corrections and modifications to improve the quality of the manuscript based on reviewers' comments. These cells are gated on CD11c and added in the revised manuscript.

  1. Are the blood glucose measurements shown in Fig 1 performed at fasting or are they fed-blood glucose levels? Please clarify.

Response: We were very grateful for the comments by the reviewers. We have taken blood glucose levels without fasting as normal blood glucose levels. We never fed extra glucose to these mice.

  1. The statement in line 522 that “ increased levels of IL-10 and IL-13…..would maintain chronic inflammation” is not substantiated. IL-10 is an immunoregulatory cytokine and its increase may be an effect to counter chronic inflammation rather than being a cause. The reviewer suggests modification of this sentence.

Response: We were very grateful for the comments and suggestions of the reviewers. We have carefully modified these sentences in the revised manuscript.  

  1. cGAS/STING are nucleic acid sensors and are not expected to increase in HFD studies. Since cGAS is higher and STING is lower in AT in panel 6D, it distracts from the main conclusion of the study and the reviewer suggests removing these.

Response: We agree with the suggestions of the reviewers. We have removed cGAS/STING data from this manuscript. 

  1. The acronym MS for metabolic syndrome is confusing because this is commonly used for multiple sclerosis.

Response: We were very grateful for the comments and modified the acronym to the full name of metabolic syndrome to avoid any confusion in this manuscript.  

  1. In the introduction section, the authors mention TH17 and Tregs repertoire in line 95. This should be changed to subsets because the repertoire for T-cells is used for T-cell receptors diversity and is misleading.

Response: We were very grateful for the comments and modified repertoire as subsets.

  1. Correct the spelling of Insulin in the Y-axis label in Fig 1D.

Response: We were very grateful for the comments and sorry for the oversight. We have modified this in the revised manuscript.

  1. In line 365, the authors state that “a few studies show IL-17 can delay the development of obesity”. This needs a citation.

Response: We were very grateful for the comments added to the citation in the revised manuscript.

Reviewer 3 Report

Sonia Kiran et al. submitted the manuscript entitled "High-fat diet-induced dysregulation of immune cells correlates 2 with macrophage phenotypes and chronic inflammation in adipose tissue". In this manuscript, they found that high fat diet induced the dysregulation of immune cells in adipose tissue and caused systemic inflammation in the obese mice. Also, modulation of IRF3/STAT3 and PPAR gamma signaling in adipose tissue were observed. This is an interesting study that may provide clues for the relationship between obesity and systemic inflammation. However, several issues can be addressed to enhance the quality of current manuscript.

  1. Authors claimed that "Dysregulation of Th17/Tregs polarizes macrophages towards M1 phenotypes in part through cGAS-IRF3-STAT3 and negatively regulating PPAR-γ mediated pathways, resulting in AT inflammation during obesity". However, most of the data are observational and they have not proved the linkage between the dysregulation of Th17/Tregs, macrophage polarization, STAT3/PPAR-γ signaling and subsequent inflammatory response. To prove what authors claimed, they should carry out some interventions to show the linkage between these elements. For example, intervention of T cells or macrophage depletion maybe adopted to show that they are crucial for the modulation of cellular signaling and systemic inflammation in obese mice. 
  2. In Fig. 5, authors claimed that Th2 response was modulated. It will be better to show the Th2 cell population also by the flow cytometry analysis.
  3. The quality of Western blot image should be improved that uneven loading is seen in the representative graphs. It will reduce the credibility of the Western blot analysis. 
  4. Authors may explain the increase of IL-10 in obese mice as it is an anti-inflammatory cytokines.

Author Response

Reply to Reviewers 3 Comments: All changes are underlined in the Text.

Sonia Kiran et al. submitted the manuscript entitled "High-fat diet-induced dysregulation of immune cells correlates 2 with macrophage phenotypes and chronic inflammation in adipose tissue". In this manuscript, they found that a high-fat diet induced the dysregulation of immune cells in adipose tissue and caused systemic inflammation in obese mice. Also, modulation of IRF3/STAT3 and PPAR gamma signaling in adipose tissue was observed. This is an interesting study that may provide clues to the relationship between obesity and systemic inflammation. However, several issues can be addressed to enhance the quality of the current manuscript.

  1. Authors claimed that "Dysregulation of Th17/Tregs polarizes macrophages towards M1 phenotypes in part through cGAS-IRF3-STAT3 and negatively regulating PPAR-γ mediated pathways, resulting in AT inflammation during obesity". However, most of the data are observational and they have not proved the linkage between the dysregulation of Th17/Tregs, macrophage polarization, STAT3/PPAR-γ signaling, and subsequent inflammatory response. To prove what the authors claimed, they should carry out some interventions to show the linkage between these elements. For example, the intervention of T cells or macrophage depletion may be adopted to show that they are crucial for the modulation of cellular signaling and systemic inflammation in obese mice. 

Response: We were very grateful for the comments and suggestions of the reviewers. We agree that some of the interventions should be used for a prudent conclusion. This is a long-term study and T cells accumulate before macrophage at 4 weeks and continued till weeks 12. The depletion of macrophages always yields conflicting results as it mediates various subsets like granulocytic and monocytic macrophages. Manipulation of T cells is in the preliminary stage and beyond the scope of this study. In our ongoing study, we are manipulating altered T cell microRNAs to see the reverse in obesity as well as signaling pathways. This will be a part of a new study and did not fit in the present study.

  1. In Fig. 5, the authors claimed that the Th2 response was modulated. It will be better to show the Th2 cell population also by the flow cytometry analysis.

Response: We were very thankful for the comments and suggestions of the reviewers. Since we did not get any changes in the Th2 population (IL-4) in any of the experimental groups by flow cytometry, thus we are diluting the Th2 claim and restricted to individual cytokines.

  1. The quality of the Western blot image should be improved so that uneven loading is seen in the representative graphs. It will reduce the credibility of the Western blot analysis. 

Response: We were very thankful for the comments and suggestions by the reviewers. However, the strong signals of actin observed in HFD fed mice are due to diet effect on adipose tissue. It is suggested in the literature that reference genes like Actin and GAPDH level varies in mice adipose tissue when they feed on HFD (Fan X, et al., 2020 Front. Nutr. 7:589771. doi: 10.3389/fnut.2020.589771). But, as suggested by the reviewers, we have repeated the western blot and have modified the best available image in the revised manuscript.

  1. Authors may explain the increase of IL-10 in obese mice as it is an anti-inflammatory cytokine

Response: We were very grateful for the comments and suggestions of the reviewers. In a previous study, it has been shown that IL-10 is upregulated in proinflammatory macrophages of obese and insulin-resistant persons ( Acosta et.al., J Clin Endocrinol Metab 104: 4552–4562, 2019).  However, IL-10 does not affect adipocyte function. In this study, the increase in systemic IL-10 levels may be from an influx of macrophages in the adipose tissue.  Data at our disposal did not warrant any speculation as we have measured the level of IL-10 only at the termination point. Time course measurement of IL-10 might suggest us better explanation as to why it increases in week 12. 

Round 2

Reviewer 3 Report

Although there is a few limitations, authors have responded to the concerns with several improvements in the manuscript. In view of the authors reply and the updated information, the manuscript is considered to be fine for publication.